# Strain-Compensated Quantum Well Asymmetric Waveguide Edge-Emitting Laser Operating at 730 nm

**DOI:** 10.3390/s25041173

**Published:** 2025-02-14

**Authors:** Lutai Fan, Lijie Cao, Peng Jia, Qian Liu, Baiheng Liu, Haofei Chen, Yongyi Chen, Li Qin, Lei Liang, Yuxin Lei, Cheng Qiu, Yue Song, Yubing Wang, Yongqiang Ning, Lijun Wang

**Affiliations:** 1State Key Laboratory of Luminescence Science and Technology, Changchun Institute of Optics, Fine Mechanics and Physics, Chinese Academy of Sciences, Changchun 130033, China; fanlutai20@mails.ucas.ac.cn (L.F.); caolijie22@mails.ucas.ac.cn (L.C.); 13756145869@163.com (B.L.); chenhaofei23@mails.ucas.ac.cn (H.C.); qinl@ciomp.ac.cn (L.Q.); liangl@ciomp.ac.cn (L.L.); leiyuxin@ciomp.ac.cn (Y.L.); qiucheng@ciomp.ac.cn (C.Q.); songyue@ciomp.ac.cn (Y.S.); wangyubing@ciomp.ac.cn (Y.W.); ningyq@ciomp.ac.cn (Y.N.); 2Daheng College, University of Chinese Academy of Sciences, Beijing 100049, China; 3State Key Laboratory of High-Power Semiconductor Lasers, Changchun University of Science and Technology, Changchun 130022, China; 4Jlight Semiconductor Technology Co., Ltd., Changchun 130102, China

**Keywords:** semiconductor lasers, high power, asymmetric waveguide, strain compensation, red lasers

## Abstract

Semiconductor lasers operating at the 730 nm peak wavelength have diverse applications, including biomedical diagnostics, agricultural lighting, and high-precision sensing. However, quantum well (QW) materials, commonly employed at this wavelength, often fail to simultaneously meet the dual requirements of lattice matching and bandgap alignment. In this study, GaAsP/AlGaInP large strain compensation QW with lattice mismatches of −7.533‰ and 1.112‰ was developed. Strain compensation was utilized to address the lattice mismatch while ensuring lasing action at 730 nm. Based on this, the impact of waveguide design, particularly graded and asymmetric waveguides, on the power output was explored. Additionally, the relationship between the doping profile of the device and lasing efficiency was investigated. The completed 100 μm wide semiconductor edge-emitting laser (EEL) achieved 730 nm continuous wave laser with 1 W output power at 2 A current. This study proposes an approach to enhance the lasing power and optoelectronic conversion efficiency of lasers and provide valuable solutions for their practical applications.

## 1. Introduction

The red semiconductor laser operating at 730 nm, known for its high efficiency, small size, and electrical pumping, has found extensive applications in photodynamic therapy [1,2], laser printing [3], material processing, and optical agriculture [4]. In optical agriculture, in particular, 730 nm lasers show great potential for complementary plant lighting [5,6]. Compared with the light-emitting diode (LED) commonly used as a plant light source, the semiconductor laser diode (LD) has obvious advantages, including narrow emission angle, strong coherence [7], high electro-optical conversion efficiency [8], and significant energy-saving effect [9], which ensure that LD can effectively promote plant growth and improve crop growth yield [10,11].

Due to the lack of suitable QW structures that matched the GaAs substrate at 730 nm [12,13], scientists had proposed various strain QW structures. As early as 1997, Smowton et al. demonstrated the feasibility of AlGaInAs/AlGaAs QWs for 730 nm laser emission [14]. In the same year, Emanuel et al. achieved 2 W continuous wave lasing at 731 nm using AlGaInAs/AlGaAs QWs [15]. Another structure is the GaAsP/AlGaAs QW, proposed by Erbert et al. [16], which achieved 7 W continuous wave output at 735 nm [17], and utilized a tapered waveguide to achieve high-beam quality lasing [18]. In 2024, Mauerhoff et al. used GaAsP/AlGaAs QWs with an asymmetric waveguide to achieve 500 mW lasing power at 725 nm and completed 1300 h reliability testing [19]. A third viable option is the InGaAsP/AlGaInP QW [20]. In 2007, Nomoto et al. used this type of QW to achieve 40 mW continuous wave lasing at 703 nm with a 2 μm ridge waveguide [21]. In 2022, Maassdorf et al. achieved 900 mW lasing power at 720 nm using InGaAsP/AlGaInP QWs [22]. In 1998, Mawst et al. demonstrated 2.9 W continuous wave output at 730 nm using InGaAsP/InGaP Al-free QWs [23,24]. In 2000, they achieved 1 W continuous wave output with over 1000 h reliability testing at 730 nm using stress-compensated InGaAsP/InGaP QWs [25].

This study proposed a novel strain-compensated GaAsP/AlGaInP QW structure to reduce the impact of lattice mismatch on the laser [26,27] power and efficiency characteristics. The findings were expected to fill the research gap on GaAsP/AlGaInP QW structures in 730 nm semiconductor lasers. In addition, this study investigated the effects of graded separation constrained heterostructure (SCH) and asymmetric waveguides on power and efficiency [28,29], along with doping profile optimization to achieve the best lasing power. The EEL of 100 μm wide ridge obtained by this work achieved 1 W continuous wave output at 730 nm under 2 A current, which provided a valuable solution for semiconductor lasers in this wavelength.

## 2. Epitaxial Structure Design

In this section, the graded SCH, asymmetric waveguide, and optimization of doping distribution were used to achieve efficient EEL design. The four EELs in this study are shown in Figure 1. EEL0 was designed in Section 2.1, which was a configuration of symmetric waveguides. EEL1 was designed in Section 2.2.1, which added a graded SCH to EEL0. EEL2 was designed in Section 2.2.2, which added an asymmetric waveguide to EEL1. EEL3, which was designed in Section 2.3, only increased the doping concentration in the waveguide compared to EEL2.

### 2.1. Active Region Design

In this study, GaAsP was employed as the well material, while (Al_0.8_Ga_0.2_)In_0.5_P was introduced on both sides as strain-compensating layers to mitigate well strain. Through the strain compensation of the well and barrier, bandgap optimization and defect reduction can be achieved in QW, which can significantly improve the photoelectric performance and stability of the device [30].

The lattice constant of (Al_0.8_Ga_0.2_)In_0.5_P is 5.6598 Å by interpolation formula. The lattice constants of GaAs, InP, AlP, and GaP, which are the basis values of the interpolation operation, are shown in Table 1. These data are from Wikipedia, and the values under 300 K are selected for calculation. According to the formula(1)$$f=\frac{a-a_{0}}{a_{0}},$$
where *a*_0_ is the lattice constant of the substrate, and *a* is the lattice constant of the material, the lattice mismatch *f* was calculated. The *f* of barrier is 1.158‰. The positive *f* value of the barrier indicates compressive strain from the substrate.

The gain spectrum of GaAsP/(Al_0.8_Ga_0.2_)In_0.5_P QW in different P components of the well material was calculated using PICS3D at an electron carrier concentration of 5 × 10^18^/cm^3^ in QW. When calculating the gain spectrum, PICS3D only needs the set electron carrier concentration to complete the calculation. The calculation results are illustrated in Figure 2. As shown in Figure 2a–e, material gain increases with the thickness of the well. This is because thicker wells can hold more charge carriers, which increases the gain by producing more photons through radiation recombination. However, well thickness cannot increase indefinitely. As the well thickens, a higher carrier density is required to maintain the gain, which results in a decrease in the peak gain. Moreover, wider wells allow high-energy carriers to be filled to higher energy levels, resulting in wavelength redshifts and spectral broadening, as observed in gain spectrum.

The reduction in P component in the well reduces the lattice constant of GaAsP, which reduces strain and increases growth stability. However, it causes wavelength to redshift from 730 nm as shown in Figure 2f. For the same emission wavelength, a thinner well requires a smaller P component to achieve the peak output. Considering both thickness and material composition, we ultimately selected a 5 nm-thick GaAsP well with a P component of 0.21 for the 730 nm edge-emitting laser (EEL). The lattice constant of GaAs_0.79_P_0.21_ was calculated as 5.6107 Å by the interpolation formula, and its lattice mismatch was −7.533% by Formula (1), which is the tensile strain. The epitaxial structure EEL0 was constructed by using AlGaAs, which is matched with GaAs, as waveguide and cladding.

The carrier concentration and energy band of the EEL0 at a current surface density of 1000 A/cm^2^ were calculated using PICS3D. These results are presented in Figure 3a. The concentration of carriers within the well was approximately two to three orders of magnitude higher than that in the other layers, indicating that nearly all carriers were injected into the QW [31]. Meanwhile, the carrier concentration within (Al_0.8_Ga_0.2_)In_0.5_P was significantly lower than that in the waveguides and well, suggesting that carriers did not overflow into the barrier layers. The separation of hole energy levels in well due to tensile strain is also shown in Figure 3a. Due to the lower mobility of the hole, its concentration in the well is lower than electron. In Figure 3a, there is an energy level deviation between Al_0.6_Ga_0.4_As and (Al_0.8_Ga_0.2_)In_0.5_P, which is due to the accuracy of energy band calculation inside the software, and will not affect the simulation results. Figure 3b illustrates that, as the concentration of carriers in QW increases, the peak of gain spectrum continues to increase. Once the gain spectrum exceeded zero, the designed device successfully achieved lasing action. Under the injected current density of 1000 A/cm^2^, the carrier concentration within the QW was found to be approximately 5 × 10^18^/cm^3^.

### 2.2. Waveguide Design

Waveguide design in an EEL is crucial for ensuring effective and efficient beam formation, confinement, and transmission. It directly affects the performance of the EEL, including the threshold current, output power, and other parameters [32,33]. By optimizing the waveguide thickness and materials, the carrier’s confinement and optical fields within the active region can be enhanced. This can subsequently improve the optical gain, thereby enabling a low-threshold lasing and higher output power. Therefore, an efficient waveguide design is critical for achieving high-performance EELs. In this paper, graded waveguide and asymmetric waveguide designs were used to improve power and efficiency.

#### 2.2.1. Graded Waveguide Design

In order to improve the optical confinement factor and carrier injection efficiency in the active region, we used AlGaAs graded SCH. The graded SCH will change the refractive index distribution to shrink the longitudinal optical field distribution, thereby increasing the optical confinement factor in the active region. At the same time, the graded SCH will change the distribution of energy bands to affect the carrier transport process. The graded SCH-EEL is EEL1, whose epitaxial structure is shown in Figure 1.

EEL0 and EEL1 with a width of 100 μm, cavity length of 2 mm, and front and rear cavity surface reflectance of 2% and 99.5% were simulated using PICS3D. The simulation results are presented in Figure 4. The Al component in waveguides varies between 0.7 and 0.5, providing an additional 0.12 eV of potential energy in Figure 4a. Under the influence of this extra potential, holes and electrons injected, respectively, from the P and N electrodes reach the QW more quickly and participate in radiative recombination. And in the case of the electron, this extra potential also inhibits its escape to the P-side. When the current is 3 A, the electron concentration in QW of EEL0 and EEL1 calculated by PICS3D were 8.29 × 10^18^/cm^3^ and 8.313 × 10^18^/cm^3^, and the hole concentration were 6.152 × 10^18^/cm^3^ and 6.915 × 10^18^/cm^3^. The increase in holes is more obvious because the mobility of holes was lower than that of electrons; therefore, the increase was more obvious.

Figure 4b shows the refractive index and vertical light field distribution of EEL0 and EEL1. Due to the change in refractive index distribution, the light field of EEL1 is more concentrated, and its QW also has a higher optical confinement factor. Through integration, the optical confinement factors of QW of EEL0 and EEL1 were 2.06% and 4.18%, respectively. The power of EEL1 was 20% higher than EEL0, and the threshold current decreased from 1.6 to 1.3 A in Figure 4c.

#### 2.2.2. Asymmetric Waveguide Design

Since the absorption cross-section of the P-side of GaAs material is larger than that of the N-side, the optical field can be shifted to the N-side by reducing the thickness of the P-side, so as to reduce the total optical loss. The optical field of EEL was simulated by COMSOL, and the optical confinement factors of each layer were calculated. The refractive index of the materials in Table 2 were derived from PICS3D at 730 nm.

Based on the following formula,(2)$$\alpha_{m}=\sum_{i}\Gamma_{i}(n_{i}*\alpha_{n}+p_{i}*\alpha_{p}),$$
the optical loss for each layer can be calculated. *Γ_i_* is the optical confinement factor, which can characterize the power of QW. *α_n_* and *α_p_* are the absorption cross-section for electrons and holes, and *n_i_* and *p_i_* are the electron and hole densities in each layer under equilibrium conditions. For GaAs-based materials, the absorption cross-section for electrons is approximately 3 × 10^−18^ cm^2^, and for holes, it is approximately 7 × 10^−18^ cm^2^ [34]. *α_m_* represents the internal optical loss in the EEL caused by doping, which directly affects the threshold current and power of the EEL.

Using EEL1 as the base device, we analyzed the influence of waveguide thickness on optical confinement factor and loss. Since EEL1 was a symmetrical graded SCH structure, the light field was centered on the QW and evenly distributed in the P- and N-waveguides on both sides. Due to the higher loss coefficient on the P-side, a symmetric waveguide is not optimal for enhancing power output. By adopting an asymmetric waveguide design and reducing the thickness of the P-side waveguide layer without changing the doping concentrations or material compositions [35], the transport distance for holes can be shortened, thereby accelerating their movement. Simultaneously, this approach lowers the average refractive index of the P-side waveguide, shifting the optical spot center toward the N-side. As a result, more light propagates through the N-side waveguide. As illustrated in Figure 5, when the P-side thickness is reduced, the optical field center gradually shifts by 300 nm from the QW toward the N-waveguide.

Figure 6a shows the trend of optical confinement factors in different regions with P-waveguide thickness. As P-waveguide was thinned, the optical confinement factor in this region decreased, while that in N-waveguide increased. The optical confinement factor was stable at 5% in the active region and below 2% in the cladding on both sides. Figure 6b shows the internal optical loss in different regions, and its variation trend is consistent with that of Figure 6a. With the thinning of the P-side waveguide, the total internal optical loss of EEL1 first decreased and then increased. At this minimum point, the thickness of the N-side waveguide was 900 nm and the thickness of the P-side waveguide was about 550 nm. EEL1 at this size had the lowest threshold current and the best power, so it is determined to be the epitaxial size of EEL2.

### 2.3. Doping Design

Doping optimization is another critical factor in designing EELs, as it directly affects crucial parameters such as carrier injection efficiency, internal loss, and electro-optical conversion efficiency [36]. Proper doping can balance the carrier injection efficiency while minimizing losses, enabling a balance between carrier transport and optical loss.

Taking the doping concentration of 5 × 10^16^/cm^3^ in P- and N-waveguides in EEL2 as the baseline, the concentration is multiplied in the waveguides. The laser power and electro-optical conversion efficiency of EEL2 with different doping concentrations of 100 μm wide, 2 mm cavity length, and surface reflectance of 2% and 99.5% at 3 A were calculated by PICS3D. In addition, the total internal optical loss of EEL2 was calculated by Equation (2). The result is shown in Figure 7. At lower doping levels, the power and electro-optical conversion efficiency of the device increased with the doping concentration, indicating that the increased of doping significantly improves the conductivity and carrier transport efficiency. With the increase in doping concentration, the carrier concentration in the active region continued to increase, and the number of carriers involved in the radiation recombination increased, thus improving the gain. As the concentration of carriers reached a certain number, the gain reached saturation. But the further increase in doping concentration in the waveguide increased the internal optical loss and decreased the gain. Combined with Figure 7, the influence of doping concentration on transport efficiency and loss was considered in a balanced manner. The doping concentration in the final designed EEL3 was 10 times of the baseline value.

## 3. Device Fabrication

The entire epitaxial structure was grown on a (100) GaAs substrate using an AIX 200/4 MOCVD system. The primary materials employed for the growth process included TMGa, TMAl, TMIn, AsH_3_, PH_3_, SiH_4_, and TMG, with H_2_ and N_2_ serving as carrier gasses to provide an optimal growth environment. To evaluate the effectiveness of the previously discussed optimization in improving EEL power and efficiency, three different epitaxial structures of EEL1, EEL2, and EEL3 were grown separately. EEL1 only includes graded SCH. EEL2 combines graded SCH and asymmetric waveguides. EEL3 includes graded SCH, asymmetric waveguides, and doping optimization. To enhance lattice quality, the substrate was initially grown with a low-temperature GaAs buffer layer. Then, N-type AlGaAs cladding and waveguide were subsequently grown, whose graded SCH were achieved through the gradual introduction of TMGa and TMAl. Next, a low-temperature GaAsP/AlGaInP strained quantum well region was deposited, effectively minimizing strain and improving interface flatness. Above the quantum well, a P-type AlGaAs waveguide layer and a cladding layer were grown, with a highly doped GaAs cap layer added as the top layer to form the ohmic contact.

Once the epitaxial layers were grown, a 100 µm wide ridge waveguide was fabricated using photolithography and etching processes. The calculations revealed that, with an etching depth of 1200 nm, a refractive index difference of 1.56 × 10^−3^ was maintained between the groove and ridge waveguide, effectively confining the optical mode within the ridge. The actual etching depth was 1220 nm, resulting in a refractive index difference of 1.57 × 10^−3^. Following the cleaving of the epitaxial layers into 2 mm cavity-length bars, they were coated with 99.5% high-reflection (HR) and 2% anti-reflection (AR) coatings to achieve single-side high-power lasing. The materials of HR were SiO_2_ and TiO_2_, and the materials of AR were TiO_2_ and Al_2_O_3_. The cavity surfaces were not passivated. The bars were then cleaved into individual EELs as shown in Figure 8, and a flip chip was bonded onto a 4 mm × 4.8 mm copper heat sink for testing.

## 4. Results and Analysis

Figure 9a,b illustrate the power and electro-optical conversion efficiency of the three epitaxial EEL structures of the single devices after coating and under continuous current excitation at 20 °C. It is evident that EEL1 and EEL2 exhibit considerably lower power and efficiency than the optimized EEL3. As shown in Figure 9a, at a current of 1 A, EEL2 generates an output power of 180 mW, compared to only 80 mW for EEL1. Additionally, EEL2 exhibits a significantly lower threshold current of 140 mA, compared to 450 mA for EEL1. These differences in the output power and threshold currents are compatible with the trends predicted by the simulation results in Figure 4c. This further confirms that introducing the graded SCH effectively barriers the lasing threshold and enhances the output power. The discrepancy observed between the simulated results (Figure 4c) and the actual device measurements (Figure 9a) arises from the idealized material properties and lasing conditions used in the simulation. In Figure 9b, the differences in the threshold current, output power, and efficiency between EEL2 and EEL3 follow the trends illustrated in Figure 7. With the optimized EEL3, stable operation was achieved at an output power of 1.5 W under a driving current of 3 A. As the current increases, the slope efficiency of EEL3 decreases gradually from 0.75 W/A at 1 A to 0.3 W/A at 3 A. At the driving current of 3 A, a power saturation trend appears due to cavity surface degradation.

The measurement and calculation method of far-field divergence angles of EEL3 at different currents were carried out according to GB/T 31359-2015 [37], which was used to characterize the beam quality and stability. The emitting surface of the EEL3 is 80 mm away from the receiving screen. The far-field beam spots on the receiving screen were measured using a 4X BEAM EXPANDER (P/N SPZ17022) (Thorlabs, Newton, NJ, USA). According to the formula(3)$$\theta =2\times arctan\frac{d}{Z},$$
where *d* is half the width of the spot, and *Z* is the distance from far-field beam spots to the emitting surface, the divergence angle *θ* was calculated. The divergence angle at 95% energy is taken as the far-field divergence angle. According to the formula(4)$$M^{2}=\frac{\pi \theta \omega }{4\lambda },$$
where *θ* is the full angle of the far-field divergence, *ω* is the waist diameter of the spot, and *λ* is the wavelength, the beam-quality factor *M*^2^ was calculated. The waist diameter of the spot on the slow axis is equivalent to the ridge waveguide width of 100 μm, while the diameter on the fast axis is the P- and N-waveguide thickness of 1.5 μm.

Figure 10a,b illustrate the variations in the slow- and fast-axis far-field divergence angles of EEL3 at different currents. In this study, the angle corresponding to 95% of the total intensity was considered as the far-field divergence angle. As displayed in Figure 10a, the slow-axis divergence angle of EEL3 is around 20°, which is large because EEL3 has a 100 μm wide stripe structure. This leads to multiple lasing modes along the slow-axis. As the current increases, more modes lase along the slow-axis direction, causing the slow-axis far-field divergence angle to increase. As illustrated in Figure 10b, the fast-axis divergence angle of EEL3 is approximately 47°. In the experiments, the waist diameter of the beam along the fast-axis direction closely matched the combined thickness of the P- and N-waveguides in the EEL, corresponding to the beam width along the longitudinal direction. In EEL3, this value is 1450 nm, significantly smaller than the waveguide width, which is approximated as the transverse waist diameter for calculating the slow-axis divergence angle. Due to optical diffraction, the smaller the waist, the larger the divergence angle. Therefore, the fast-axis far-field divergence angle is larger than the slow-axis divergence angle.

*M*^2^ was calculated to better characterize the beam quality and investigate the impact of current variations on the modes of the slow and fast axes. The calculated *M*^2^ values corresponding to the divergence angles at different currents exhibit a trend consistent with the changes in the divergence angle, as illustrated in Figure 11. Although EEL3 exhibits a large divergence angle on the fast axis, its *M*^2^ remains stable within the range of 1.3 to 1.34, indicating single-mode excitation characteristics along the fast axis. However, *M*^2^ along the slow axis is approximately 40, suggesting that multiple modes are simultaneously lasing in the transverse direction. This is unfavorable for high-precision device applications. In future studies, single-mode lasing in the slow-axis direction could be achieved by reducing the waveguide width.

The temperature-dependent characteristics of EEL3 were measured to assess the temperature stability and reliability of the device. As displayed in Figure 12a,b, as the temperature increases, the output power of the device decreases and the threshold current increases. This is primarily due to the enhanced rate of thermal carrier escape at high temperatures. Additionally, at elevated temperatures, the large strains in the GaAsP quantum well and AlGaInP barriers experience greater thermal expansion and strain mismatch, further contributing to the reduction in output power and efficiency. As illustrated in Figure 12a, EEL3 maintains an output power of 1 W at 50 °C. Figure 12c illustrates the temperature-dependent spectrum of EEL3 at 2 A. The device achieves a full width at half maximum (FWHM) of 2 nm at the peak wavelength of 730 nm when operated at 15 °C. As the temperature increases, the bandgap of the QW gradually widens, causing the wavelength to redshift at a rate of approximately 0.27 nm/°C. When the temperature reaches 75 °C, the spectral gain decreases significantly, and by 80 °C, no distinct lasing peak is observed. It is worth mentioning that the power degradation of EEL3 under high-temperature conditions is reversible. Moreover, when the temperature decreases, the device can return to its normal operating state.

## 5. Discussion

For the practical applications of the 730 nm EEL, both high output power and efficiency are essential, along with minimizing the far-field divergence angle. First, the higher output power and efficiency ensure that the device meets the application demands while minimizing power consumption. Second, a smaller far-field divergence angle results in better beam quality, which is beneficial for precise laser operations such as laser scalpels. In some applications, higher demands are placed on the temperature-dependent characteristics of EELs. For example, in agricultural lighting applications, greenhouses often operate at temperatures ranging from 40 °C to 60 °C throughout the year. Due to space constraints, installing cooling systems in such environments is generally not feasible. Therefore, when designing a laser, it is necessary to account for the wavelength redshift and power degradation caused by high environmental temperatures.

Based on the work summarized in Section 1, many mature 730 nm EEL designs already existed. However, the GaAsP/AlGaInP strain compensation design in this study had not been previously reported due to the challenges of growing these materials at high strains. This has certain requirements for future research. To achieve higher power and efficiency, future research will focus on the optimization of QW and waveguide, such as the study of the optimal lattice mismatch of AlGaInP barriers to achieve the best strain compensation effect, or the research on the best growth conditions for GaAsP/AlGaInP QW. These efforts are expected to significantly improve output power characteristics.

## 6. Conclusions

In this study, we developed a GaAsP/AlGaInP strain-compensated semiconductor red laser with a peak wavelength of 730 nm. Through the optimization of a graded SCH, asymmetric waveguide design, and doping optimization, the device achieved an output power of 1.5 W under a continuous driving current of 3 A, with a cavity length of 2 mm and a ridge waveguide width of 100 µm. The highest electro-optical conversion efficiency was 20% at room temperature. The device achieved far-field divergence angles of 47° and 20° along the fast and slow axis, respectively. Using M^2^ for characterizing the beam quality, single-mode emission with *M*^2^ < 2 was obtained in the fast-axis direction. Future research will focus on using a narrow-ridge structure to shrink the slow-axis beam and achieve single-mode emissions in the slow-axis direction. This device also ensured a stable output of over 1 W at 50 °C, providing practical application value for semiconductor lighting in agriculture.

## Figures and Tables

**Figure 1 sensors-25-01173-f001:**

The four types of EELs demonstrate the research process in this study, where PCld is the P-side cladding, PWG is the P-side waveguide, NWG is the N-side waveguide, and NCld is the N-side cladding.

**Figure 2 sensors-25-01173-f002:**
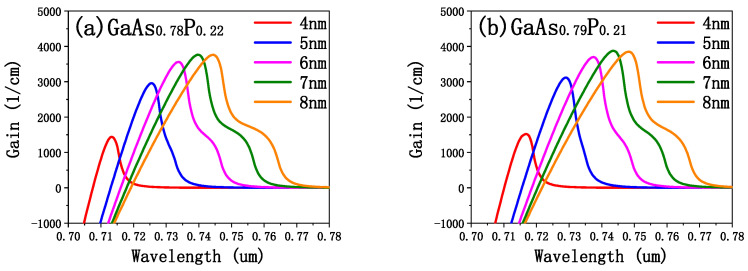
(**a**–**e**) Gain spectrum of GaAsP/(Al_0.8_Ga_0.2_)In_0.5_P QW with different P compositions of GaAsP at various thicknesses of well; (**f**) the wavelength variation corresponding to the gain peak of the gain spectrums in (**a**–**e**).

**Figure 3 sensors-25-01173-f003:**
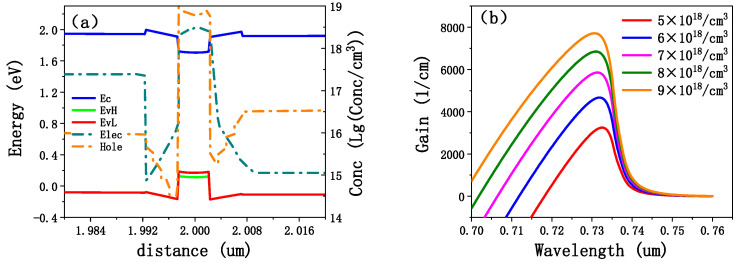
(**a**) Band diagram and electron and hole concentration distributions at a current density of 1000 A/cm^2^, where Ec is the valence band bottom level, EvH is the heavy hole level, EvL is the light hole level, Elec is the electron concentration, and hole is the hole concentration; (**b**) gain spectrum corresponding to different carrier (electron) concentrations in QW.

**Figure 4 sensors-25-01173-f004:**
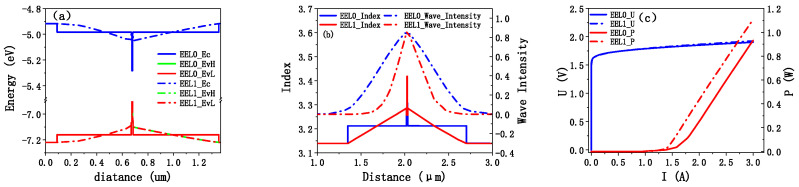
(**a**) Band diagram comparison of EEL0 and EEL1, where Ec is the valence band bottom, EvH is the heavy hole level, and EvL is the light hole level; (**b**) the refractive index distribution of EEL0 and EEL1 and the optical field distribution in the vertical direction, and Wave_Intensity indicates the optical field intensity; (**c**) simulation current–voltage curve (U) and current–power curve (P) comparison of EEL0 and EEL1.

**Figure 5 sensors-25-01173-f005:**
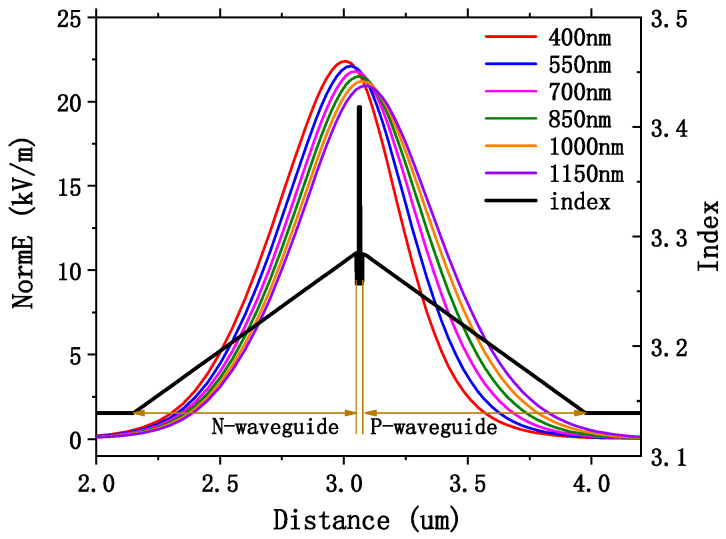
Refractive index and mode distribution for the symmetric waveguide and mode distribution of EEL1 for different P-side waveguide thicknesses. The NormE is the intensity of the light field.

**Figure 6 sensors-25-01173-f006:**
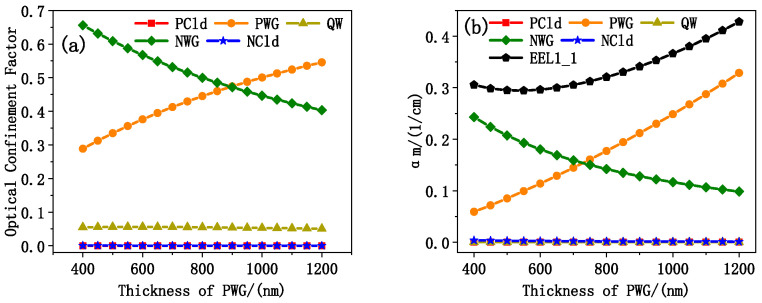
(**a**) The optical confinement factor of different regions of EEL1 varies with the thickness of PWG, where PCld is P-cladding, PWG is P-waveguide, QW is active region, NWG is N-waveguide, and NCld is N-cladding; (**b**) curves of *α_m_* in different regions of EEL1 with the thickness of PWG, where the curves indicated by EEL1 represent the *α_m_* of the entire device.

**Figure 7 sensors-25-01173-f007:**
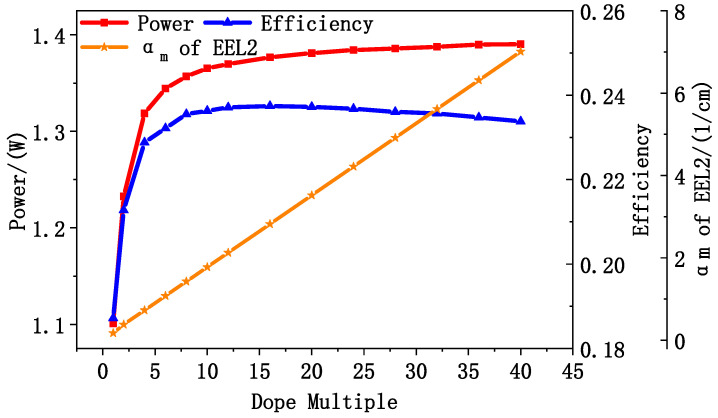
Lasing power and electro-optical conversion efficiency variation curves of the EEL2 at different doping multiples, 3 A, calculated by PICS3D. Internal optical loss calculated by Formula (2).

**Figure 8 sensors-25-01173-f008:**
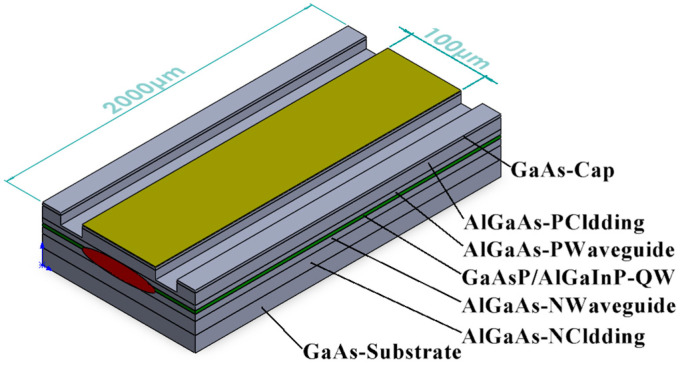
Schematic diagram of EELs.

**Figure 9 sensors-25-01173-f009:**
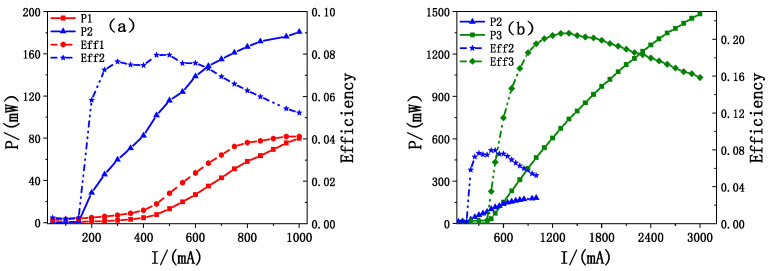
(**a**) The current-power (P1, P2) and current-efficiency (Eff1, Eff2) curves of EEL1 and EEL2; (**b**) The current-power (P2, P3) and current-efficiency (Eff2, Eff3) curves of EEL2 and EEL3.

**Figure 10 sensors-25-01173-f010:**
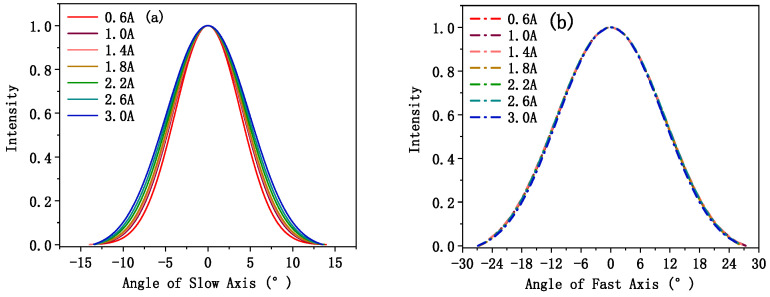
Far-field images and divergence angles of (**a**) the slow axis and (**b**) the fast axis of EEL3 under different currents at 20 °C.

**Figure 11 sensors-25-01173-f011:**
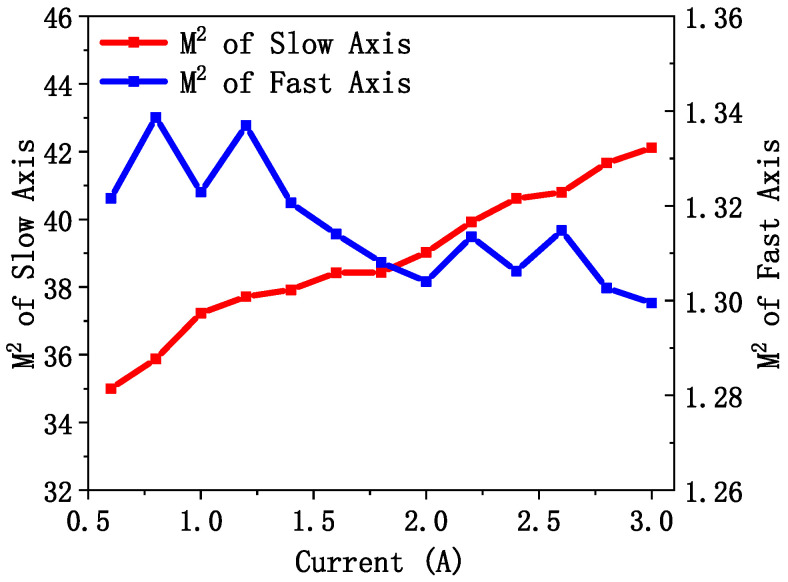
*M*^2^ of the slow axis and fast axis of EEL3 under different currents at 20 °C.

**Figure 12 sensors-25-01173-f012:**
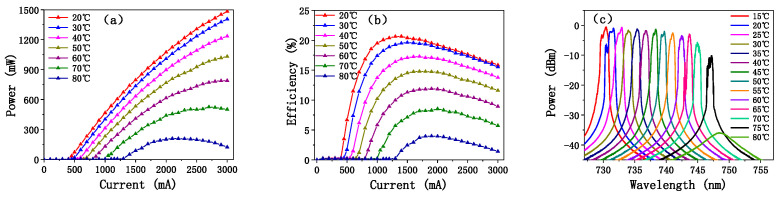
(**a**) Power, (**b**) efficiency, and (**c**) spectrum of EEL3 at different temperatures (CW).

**Table 1 sensors-25-01173-t001:** Crystal lattice constants of each material at 300 K.

Material	Lattice Constant/Å
GaAs	5.6533
InP	5.8687
AlP	5.451
GaP	5.447

**Table 2 sensors-25-01173-t002:** Refractive index of the material at 730 nm.

Material	Refractive Index
GaAs	3.65
GaAs_0.79_P_0.21_	3.418
Al_0.7_GaAs	3.139
Al_0.6_GaAs	3.212
Al_0.5_GaAs	3.285
(Al_0.8_Ga_0.2_)In_0.5_P	3.257

## Data Availability

Data are contained within the article.

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
