# Peer review of "Strain-Compensated Quantum Well Asymmetric Waveguide Edge-Emitting Laser Operating at 730 nm"

_sensors, 2025, doi:10.3390/s25041173_

Round 1

Reviewer 1 Report

Comments and Suggestions for Authors

The manuscript focuses on the development of semiconductor lasers operating at a wavelength of 730 nm. While the authors present results from calculations and experiments, further clarifications and enhancements are necessary. Below are some of my comments.

In the abstract, please specify the tensile misfit when discussing the large-stress-compensated quantum well (QW), the stripe width, and the current at which 1 W is achieved. Additionally, use “strain” instead of “stress” throughout the text.

Lines 27-28: Instead of “doping concentration,” the term “doping profile” would be more appropriate in this context.

The introduction contains unnecessary comparisons between LED sources and semiconductor laser diodes, several sentences about the application of lasers in agriculture, and the ambiguous term “photosynthetic compensation point”. Furthermore, the authors use the non-scientific term “foreign studies” and justify the relevance of the work by stating “to address the research gap in 730 nm semiconductor lasers in China,” which is not suitable.

More references from the last 10 years on the design and light-current curves of 730 nm lasers should be included.

To support the claim that AlGaInP reduces the barriers in the well, the authors could provide XRD results or a comparison of the laser characteristics of two structures differing by the presence of AlGaInP layers.

To evaluate the calculations conducted and to increase the overall value of the work, a Table summarizing all significant parameters of the simulation model is necessary. In the current version of the manuscript, it is challenging to clearly distinguish the structures compared by the authors in calculations and experiments. The authors should also add Tables detailing the compositions, thicknesses, doping profiles of all layers, the barrier heights, QW depth (defined by the energy difference between the first electron level in the QW and the conduction band edge of the surrounding layers), and the optical confinement factor for all calculated and experimental structure designs.

A significant question arises from the fact that a structure without gradients and without AlGaInP layers is used in the study [https://doi.org/10.1109/JSTQE.2024.3431225 Mauerhoff, Felix, et al. "GaAs Based Edge Emitters at 626 nm, 725 nm and 1180 nm." IEEE Journal of Selected Topics in Quantum Electronics], however laser characteristics are similar to those of the authors’ work, such as slope efficiency and threshold. Can the authors explain this?

Line 96: The phrase “low refractive index” requires clarification—compared to which layer does the AlGaAs layer have a lower refractive index?

Line 105 and throughout the text: please use A/cm² for current density instead of A/mm².

In Figure 2(a), it is evident that the electron and hole concentrations in the quantum well are not equal. How can the authors comment on this?

Can the authors explain how they envision the application of “escape process can be suppressed by applying the reverse additional potential to the opposite side”?

Lines 163-164: It would be beneficial to add references for the optical loss coefficients.

Figure 5: Please replace “Limitefactor” with “Optical confinement factor.”

Can the authors specify in the figure captions or legends which designs of laser heterostructures they refer to?

Can the authors elucidate the physical picture of the process described in line 225, “carrier recombination in the active region becomes saturated,” and line 226, “increase the loss during the transport process”?

Line 228: “baseline value.” Can the authors provide this value?

Line 228: “Higher doping concentrations enhance the carrier transport efficiency.” Can the authors provide the concentration value at which enhanced carrier transport efficiency is observed?

Line 236: The figure number needs to be changed. It is unclear at what pumping current the curves shown in this figure were obtained.

Line 243: “three distinct” repeats “three different.”

Line 278: The figure number needs to be changed.

Line 288: The authors state that a 100 μm wide stripe structure leads to multiple lasing modes along the slow axis. However, these modes do not manifest in Figure 8(a). How do the authors account for this?

Can the authors specify the coating material on the mirrors and whether passivation was applied before the coatings were deposited?

Please improve the English in phrases such as line 23 “inefficiencies in QWs performance,” line 60 “reasonable optimization,” line 62 “doping treatment,” line 133 “suppress the transmission of opposite-polarity carriers,” line 201 “causing the overall reduction in the P-side loss to gradually saturate or even increase,” and line 246 “nonuniform waveguide design.”

In some references, phrases like “[Page Range if Available]” appear. For reference 24, the necessary data is missing.

Comments on the Quality of English Language

Please improve the English in phrases such as line 23 “inefficiencies in QWs performance,” line 60 “reasonable optimization,” line 62 “doping treatment,” line 133 “suppress the transmission of opposite-polarity carriers,” line 201 “causing the overall reduction in the P-side loss to gradually saturate or even increase,” and line 246 “nonuniform waveguide design.”

Author Response

Comments 1: [In the abstract, please specify the tensile misfit when discussing the large-stress-compensated quantum well (QW), the stripe width, and the current at which 1 W is achieved. Additionally, use “strain” instead of “stress” throughout the text.]

Response 1: [Thank you for pointing those out. We agree with those comments. Therefore, we have made three changes. First, we add a description of the lattice mismatches of large strain QW in the abstract section, in line 24. Second, we have added a description of the size and current state of the EEL when it is implemented at 1W, in line 30. Third, as you suggested, we use "strain" instead of "stress."]

Comments 2: [Lines 27-28: Instead of “doping concentration,” the term “doping profile” would be more appropriate in this context.]

Response 2: [Thank you for pointing this out. We agree with this comment. We have replaced "doping concentration" with "doping profile" on line 27.]

Comments 3: [The introduction contains unnecessary comparisons between LED sources and semiconductor laser diodes, several sentences about the application of lasers in agriculture, and the ambiguous term “photosynthetic compensation point”. Furthermore, the authors use the non-scientific term “foreign studies” and justify the relevance of the work by stating “to address the research gap in 730 nm semiconductor lasers in China,” which is not suitable.]

Response 3: [Thank you for pointing those out. We agree with those comments. Therefore, we have made four changes. First, we delete most of the comparison between LED and LD, but because the practical application field and practical value of this study are biased toward agricultural lighting, only some advantages of LD in agriculture are left to describe. Secondly, we have deleted all agricultural terms. Third, we have changed the presentation of existing studies and added references to them, between lines 46 and 60. Finally, we redefined the research meaning of this manuscript as “proposes a novel strain-compensated GaAsP/AlGaInP QW structure” (line 62) and “provides a valuable solution for semiconductor lasers in this wavelength” (line 69).]

Comments 4: [More references from the last 10 years on the design and light-current curves of 730 nm lasers should be included.]

Response 4: [Thank you for pointing those out. We agree with those comments. Therefore, we have updated the content of the introduction section to include the addition of nearly 30 years of research progress and the existing results of various strain QW structures, between lines 46 and 60.]

Comments 5: [To support the claim that AlGaInP reduces the barriers in the well, the authors could provide XRD results or a comparison of the laser characteristics of two structures differing by the presence of AlGaInP layers.]

Response 5: [Thank you for your advice. However, we did not mention in the manuscript that "AlGaInP can reduce the barrier inside the well." Perhaps our lack of clarity led to misunderstanding, so we marked the components of each material in AlGaInP and re-changed some sentences that were not clear.]

Comments 6: [To evaluate the calculations conducted and to increase the overall value of the work, a Table summarizing all significant parameters of the simulation model is necessary. In the current version of the manuscript, it is challenging to clearly distinguish the structures compared by the authors in calculations and experiments. The authors should also add Tables detailing the compositions, thicknesses, doping profiles of all layers, the barrier heights, QW depth (defined by the energy difference between the first electron level in the QW and the conduction band edge of the surrounding layers), and the optical confinement factor for all calculated and experimental structure designs.]

Response 6: [Thank you for pointing those out. We agree with those comments. Therefore, we made the following changes. Firstly, Figure 1 is added to illustrate the epitaxial structure of the devices involved in the manuscript. If there are certain parameters involved in the research process, we have given them. Secondly, we have explained where optical limiting factors need to be used. Third, for the calculation, all the values we can provide have been given. However, the calculations involving energy band and power are carried out using PICS3D, so we cannot give the internal values of the software. If it is the value we set, it is already given.]

Comments 7: [A significant question arises from the fact that a structure without gradients and without AlGaInP layers is used in the study [https://doi.org/10.1109/JSTQE.2024.3431225 Mauerhoff, Felix, et al. "GaAs Based Edge Emitters at 626 nm, 725 nm and 1180 nm." IEEE Journal of Selected Topics in Quantum Electronics], however laser characteristics are similar to those of the authors’ work, such as slope efficiency and threshold. Can the authors explain this?]

Response 7: [Thank you for pointing those out. In this manuscript, we only propose the EEL3 structure of an asymmetric graded waveguide GaAsP/AlGaInP quantum well, and also prepare EEL1 and EEL2, and discuss the lasing characteristics of the three EEL types. Finally, we believe that the graded SCH and asymmetric waveguide can effectively improve the efficiency. This is evident in the comparison of the effects of the three structures in this manuscript. At the same time, we briefly discuss the devices in the manuscript in the fifth part, and believe that the performance of this device has a certain room for improvement.]

Comments 8: [Line 96: The phrase “low refractive index” requires clarification—compared to which layer does the AlGaAs layer have a lower refractive index?]

Response 8: [Thank you for pointing this out. We agree with this comment. Therefore, we made a change. We change the expression of this sentence to “The epitaxial structure EEL0 was constructed by using AlGaAs, which is matched with GaAs, as waveguide and cladding”, in line 132.]

Comments 9: [Line 105 and throughout the text: please use A/cm² for current density instead of A/mm².]

Response 9: [Thank you for pointing this out. We agree with this comment. Therefore, we made a change. We have changed them throughout the article.]

Comments 10: [In Figure 2(a), it is evident that the electron and hole concentrations in the quantum well are not equal. How can the authors comment on this?]

Response 10: [Thank you for pointing this out. Please allow us to explain: because the effective mass of the hole is larger and the migration rate is lower, the carrier concentration of the hole in QW is usually lower.]

Comments 11: [Can the authors explain how they envision the application of “escape process can be suppressed by applying the reverse additional potential to the opposite side”?]

Response 11: [Thank you for pointing this out. Please allow us to explain: taking electrons as an example, after injection from the N side, there will be a promoting negative potential to accelerate the injection of electrons into QW; when electrons escape from QW, there is a positive potential on the P side to inhibit electron migration to the P electrode.]

Comments 12: [Lines 163-164: It would be beneficial to add references for the optical loss coefficients.]

Response 12: [Thank you for pointing this out. We agree with this comment. We give this reference [34]. It should be noted that the optical loss coefficient mentioned in the literature is at the wavelength of 1um, but the change trend is consistent.]

Comments 13: [Figure 5: Please replace “Limitefactor” with “Optical confinement factor.”]

Response 13: [Thank you for pointing this out. We agree with this comment. We have completed the replacement in Figure 5.]

Comments 14: [Can the authors specify in the figure captions or legends which designs of laser heterostructures they refer to?]

Response 14: [Thank you for pointing this out. We agree with this comment. We have marked in all the figures which structure it belongs to and added tables to show the composition of these structures.]

Comments 15: [Can the authors elucidate the physical picture of the process described in line 225, “carrier recombination in the active region becomes saturated,” and line 226, “increase the loss during the transport process”?]

Response 15: [Thank you for pointing this out. We agree with this comment. We have revised “However, at higher doping multiples, carrier recombination in the active region be-comes saturated, and additional impurities no longer facilitate carrier transport” to “With the increase of doping concentration, the carrier concentration in the active region continues to increase, and the number of carriers involved in the radiation recombination increases, thus improving the gain. As the concentration of carriers reaches a certain number, the gain reaches saturation. But the further increase of doping concentration in the waveguide will increase the internal optical loss and decrease the gain”, in line 260.]

Comments 16: [Line 228: “baseline value.” Can the authors provide this value?]

Response 16: [Thank you for pointing this out. We agree with this comment. We have improved the presentation in this section and explained the source of the base values on line 252.]

Comments 17: [Line 228: “Higher doping concentrations enhance the carrier transport efficiency.” Can the authors provide the concentration value at which enhanced carrier transport efficiency is observed?]

Response 17: [Thank you for pointing this out. We agree with this comment. We have updated the wording of this paragraph and improved the presentation of the data. We added the base value in line 252. In Section 2.3, as long as the doping in P and N waveguides is multiplied, the doping concentration in P and N waveguides at any point in Figure 7 can be calculated according to the reference value, and the two are numerically consistent.]

Comments 18: [Line 236: The figure number needs to be changed. It is unclear at what pumping current the curves shown in this figure were obtained.]

Response 18: [Thank you for pointing this out. We agree with this comment. We changed the figure number, updated the presentation of the data, and provided the current in the Figure 6.]

Comments 19: [Line 243: “three distinct” repeats “three different.”]

Response 19: [Thank you for pointing this out. We agree with this comment. We change “To evaluate the effectiveness of the optimizations discussed earlier in enhancing the power and efficiency of the EEL, three distinct three different epitaxial structures were developed” to “To evaluate the effectiveness of the previously discussed optimization in improving EEL power and efficiency,  three different epitaxial structures of EEL1, EEL2 and EEL3 were grown separately” in line 275.]

Comments 20: [Line 278: The figure number needs to be changed.]

Response 20: [Thank you for pointing this out. We agree with this comment. The figure number has been changed in line 320.]

Comments 21: [Line 288: The authors state that a 100 μm wide stripe structure leads to multiple lasing modes along the slow axis. However, these modes do not manifest in Figure 8(a). How do the authors account for this?]

Response 21: [Thank you for pointing this out. We agree with this comment. Please allow us to explain: the far-field divergence angle figure in Figure 10 was measured on the observation screen 80mm away from the EEL emission surface and after the slow-axis laser mode was transmitted in space, the light field energy superimposed on each other presented an envelope curve.]

Comments 22: [Can the authors specify the coating material on the mirrors and whether passivation was applied before the coatings were deposited?]

Response 22: [Thank you for pointing this out. We agree with this comment. We added the specification on line 294: “The materials of HR were SiO2 and TiO2, and the materials of AR were TiO2 and Al2O3. The cavity surfaces were not passivated.”]

Comments 23: [Thank you for pointing this out. We agree with this comment. We have made a lot of revisions to the manuscript, all the expressions you pointed out are involved, you can find the improved expressions in the following lines respectively: line 23 (delete); line 65 (replaced by “this study investigated the effects of graded separation constrained heterostructure (SCH) and asymmetric waveguides on power and efficiency”); line 67(replaced by “doping profile”); line 177(replaced by “And in the case of the electron, this extra potential also inhibits its escape to the P-side.”); line 237(replaced by “With the thinning of the P-side waveguide, the total internal optical loss of EEL1_1 first decreased and then increased.”); line 279(replaced by “asymmetric waveguides”).]

Comments 24: [In some references, phrases like “[Page Range if Available]” appear. For reference 24, the necessary data is missing.]

Response 24: [Thank you for pointing this out. We agree with this comment. We added the reference data in line 481, 489, 491, 495]

Reviewer 2 Report

Comments and Suggestions for Authors

The authors demonstrated an asymmetric waveguide diode laser operating at 730 nm. As the topic is of interest to the community, I’d suggest the paper to be published. Before publication, however, there are some small issues that need consideration:

1、  It is advised to include a more in-depth explanation of the growth method used for the GaAsP/AlGaInP strain-compensated quantum wells in Chapter 3 of the manuscript.

2、  In Chapter 5 of the manuscript, titled "discussion," the authors outline numerous ways to enhance the progression of this research. It is advisable to concentrate on the main method.

3、  In the explanation of Figure 5 (lines 186 to 206), the authors state, "The increase in doping loss in the P cladding layer gradually exceeded the reduction in loss in the P waveguide, causing the overall reduction in the P-side loss to gradually saturate or even increase" (lines 199 to 202). However, I did not find related information about the loss of each layer. Therefore, I suggest adding the curves of the P- and N- cladding layer losses as a function of P-waveguide thickness.

Comments on the Quality of English Language

The paper has a lot of linguistic errors needing a competent help. The English needs to be reviewed and properly edited for the use of plurality, grammar, and meaning. In addition, I would like to offer several recommendations regarding the revision of the expressions in the manuscript:

(1) "One can observe from Figure 2(a) that there is an energy band offset of approximately 0.05 eV between the AlGaInP barrier and the AlGaAs waveguide, which is due to errors in the energy band calculation process" (lines 110-112). This sentence requires further clarification, particularly regarding the additional "0.05 eV." Does this value influence the simulation and experimental results? If so, what specific effects does it have on these results?

 (2) "Due to the unequal optical loss coefficients on the P and N sides of the EEL, it is common to reduce the thickness of one side. This adjustment causes the optical field to shift toward the side with the lower loss coefficient, thereby reducing overall optical losses, provided that doping levels on both sides remain similar" (lines 152-156). The fundamental reason for the change in the optical field by tailoring the refractive index distribution should be added here.

(3) It is suggested to modify "As displayed in Figure 8(a), the slow-axis divergence angle of EEL3 is around 20°, which is large because EEL3 has a 100 μm wide stripe structure. This leads to multiple lasing modes along the slow-axis" (lines 287-289). The divergence angle is much larger than conventional diode laser with similar stripe width. This needs more explanation.

Author Response

Comments 1: [It is advised to include a more in-depth explanation of the growth method used for the GaAsP/AlGaInP strain-compensated quantum wells in Chapter 3 of the manuscript.]

Response 1: [Thank you for pointing those out. We agree with those comments. We provide as much detail as possible on line 279, however other details such as the specific growth temperature and the ratio of individual gases are related to commercial products and therefore cannot be disclosed.]

Comments 2: [In Chapter 5 of the manuscript, titled "discussion," the authors outline numerous ways to enhance the progression of this research. It is advisable to concentrate on the main method.]

Response 2: [Thank you for pointing those out. We agree with those comments. We add more specific follow-up details in line 393.]

Comments 3: [In the explanation of Figure 5 (lines 186 to 206), the authors state, "The increase in doping loss in the P cladding layer gradually exceeded the reduction in loss in the P waveguide, causing the overall reduction in the P-side loss to gradually saturate or even increase" (lines 199 to 202). However, I did not find related information about the loss of each layer. Therefore, I suggest adding the curves of the P- and N- cladding layer losses as a function of P-waveguide thickness.]

Response 3: [Thank you for pointing those out. We agree with those comments. We have changed the presentation of this section from lines 232 to 241. Optical limiting factor and loss data for each layer are added in Figure 6.]

Comments 4: ["One can observe from Figure 2(a) that there is an energy band offset of approximately 0.05 eV between the AlGaInP barrier and the AlGaAs waveguide, which is due to errors in the energy band calculation process" (lines 110-112). This sentence requires further clarification, particularly regarding the additional "0.05 eV." Does this value influence the simulation and experimental results? If so, what specific effects does it have on these results?]

Response 4: [Thank you for pointing those out. We agree with those comments. We added “which is due to the accuracy of energy band calculation inside the software, and will not affect the simulation results” on line 143 to clarify the issue.]

Comments 5: ["Due to the unequal optical loss coefficients on the P and N sides of the EEL, it is common to reduce the thickness of one side. This adjustment causes the optical field to shift toward the side with the lower loss coefficient, thereby reducing overall optical losses, provided that doping levels on both sides remain similar" (lines 152-156). The fundamental reason for the change in the optical field by tailoring the refractive index distribution should be added here.]

Response 5: [Thank you for pointing those out. We agree with those comments. We added "this approach lowers the average refractive index of the P-side waveguide, shifting the optical spot center toward the N-side." in line 223 to explain the fundamental reason why the light field distribution can be changed after the thickness of the waveguide is changed.]

Comments 6: [It is suggested to modify "As displayed in Figure 8(a), the slow-axis divergence angle of EEL3 is around 20°, which is large because EEL3 has a 100 μm wide stripe structure. This leads to multiple lasing modes along the slow-axis" (lines 287-289). The divergence angle is much larger than conventional diode laser with similar stripe width. This needs more explanation.]

Response 6: [Thank you for pointing those out. Your question is very relevant. However, for the large divergence angle of the slow axis in the manuscript, we can explain that the 100um wide ridge causes multiple modes to lasing together on the slow axis. We can reduce the slow-axis divergence angle by narrowing the ridge waveguide width until the single transverse mode cutoff condition is achieved, which is also the research process of realizing the single transverse mode lasing and our subsequent research.]

Reviewer 3 Report

Comments and Suggestions for Authors

The work concerns the design and manufacture of semiconductor edge-emitting lasers (EEL) operating at a wavelength of 730 nm. Such lasers, emitting powers ranging from a few to several tens of milliwatts, and even reaching 1000 mW or 1500 mW, with both continuous wave (CW) and pulsed operation, are commercially available, as acknowledged by the authors themselves. However, the information contained in this work may be useful for local research and technology groups, as also suggested by the authors: “Our findings are expected to address the research gap in 730 nm semiconductor lasers in China and provide guidance for practical applications of these devices.” It is also worth noting that the reviewed work combines theoretical and experimental studies, which justifies its publication. Nevertheless, before publication, the authors should address certain errors and inaccuracies.

  1. In lines 37 and 38, the authors write: “The red semiconductor laser operating at 730 nm, known for its excellent monochromaticity …”. What exactly do the authors mean here? Many good things can be said about edge-emitting semiconductor lasers, but the beam quality parameters (especially monochromaticity), compared to other types of lasers such as gas lasers, leave much to be desired.
  2. The work is largely theoretical. Proper understanding of the study requires a clear comprehension of the assumptions under which calculations were performed. Therefore, the work should include illustrative diagrams of the considered structure along with its basic dimensions, as well as a detailed table describing the individual laser layers, including material composition, thickness, doping levels, etc.
  3. In Figures 1, 2, and 5, where the gain is presented in units of amplification, the “-1” should be written as a superscript or alternatively expressed as 1/cm.
  4. Figure 1 shows the design of the active region composition to achieve optimal conditions for 730 nm wave emission. Unfortunately, it seems that this analysis was likely performed for a temperature of 300 K. However, the gain may shift significantly as the temperature of the active region increases, especially since the discussed lasers emit over 1 W of optical power. Can the authors comment on this point? Will the selected composition of the active region also be suitable for higher temperatures?
  5. In lines 104 and 105, the authors write: “Under an injected current density 104 of 10 A/mm², the concentration of the photogenerated carriers within the QW layer …”. What type of photogeneration of carriers is being referred to? Could this possibly pertain to the carrier concentration in the QW? Similarly, the text in lines 116 and 117 is unclear.
  6. Figure 2 presents the results of gain calculations. Firstly, the authors omit the effect of temperature, as inferred from the figure itself. Secondly, what losses do the authors assume when converting carrier concentration to gain? For example, how are losses represented in the ABC model? Unfortunately, the brief mention that the calculations are performed using the kp model provides no clarity.
  7. What exactly does the statement in lines 156 and 157 mean: “Using finite element analysis, optical field simulations were conducted on the designed 730 nm EEL …”? FEM is a numerical method for solving partial differential equations. Such a description does not clarify what the authors specifically did.
  8. Concerning Figures 4 and 5, the authors should describe the axes and labels on the figures more carefully. Abbreviations or concatenated words should not be used as labels unless their meaning is explained in the figure captions.
  9. Comparing Figure 5, where gain is given in units of 1/cm, with Equation (1) and the text in lines 163 and 164, it seems there is an issue with the units. What do losses of 0.1 1/cm represent? Are these material losses or modal losses? The authors should create a table listing the individual laser layers, along with their absorption losses and refractive indices.
  10. What does the statement in lines 230 and 231 mean: “… a doping concentration five times the baseline value was selected for the final 230 design”? Unfortunately, the baseline value of the concentration does not appear to be provided in the work. This confirms the need for a table as mentioned in point 2.
  11. In line 236, it says Figure 1. This is an error. It should be Figure 6. Furthermore, the vertical axes on this figure lack labels. How do the losses shown in this figure relate to the losses mentioned earlier in the text? Are the loss values appropriate?
  12. In line 278, it says Figure 2. It should be Figure 7. Additionally, the figure caption should refer to both part (a) and part (b) of the figure.
  13. Were the plots in Figure 10 generated for continuous wave (CW) operation or pulsed operation? Presumably, plots (a) and (b) were generated for CW operation. For what conditions was plot (c) generated? If it is also for CW operation, how did the authors estimate the wavelength shift with temperature? What can the authors say about the temperature in the laser’s active region during CW operation, emitting approximately 1 W of optical power?

Author Response

Comments 1: [In lines 37 and 38, the authors write: “The red semiconductor laser operating at 730 nm, known for its excellent monochromaticity …”. What exactly do the authors mean here? Many good things can be said about edge-emitting semiconductor lasers, but the beam quality parameters (especially monochromaticity), compared to other types of lasers such as gas lasers, leave much to be desired.]

Response 1: [Thank you for pointing this out. We agree with this comment. We changed “The red semiconductor laser operating at 730 nm, known for its excellent mono-chromaticity, coherence, and brightness” to "high efficiency, small size and electrical pumping" on line 37.]

Comments 2: [The work is largely theoretical. Proper understanding of the study requires a clear comprehension of the assumptions under which calculations were performed. Therefore, the work should include illustrative diagrams of the considered structure along with its basic dimensions, as well as a detailed table describing the individual laser layers, including material composition, thickness, doping levels, etc.]

Response 2: [Thank you for pointing this out. We agree with this comment. We have attached Figure 1 and 7 to illustrate the epitaxial structure of various EELs.]

Comments 3: [In Figures 1, 2, and 5, where the gain is presented in units of amplification, the “-1” should be written as a superscript or alternatively expressed as 1/cm.]

Response 3: [Thank you for pointing this out. We agree with this comment. We have changed the unit of gain and loss to "1/cm" in Figure 2,3,6,7.]

Comments 4: [Figure 1 shows the design of the active region composition to achieve optimal conditions for 730 nm wave emission. Unfortunately, it seems that this analysis was likely performed for a temperature of 300 K. However, the gain may shift significantly as the temperature of the active region increases, especially since the discussed lasers emit over 1 W of optical power. Can the authors comment on this point? Will the selected composition of the active region also be suitable for higher temperatures?]

Response 4: [Thank you for pointing this out. Allow us to explain. During the design process, we mainly focused on the power and efficiency characteristics of the device, and did not conduct detailed simulation of the temperature characteristics of the device. In the course of the experiment, considering the subsequent application scenarios, the variable temperature test was carried out.]

Comments 5: [In lines 104 and 105, the authors write: “Under an injected current density 104 of 10 A/mm², the concentration of the photogenerated carriers within the QW layer …”. What type of photogeneration of carriers is being referred to? Could this possibly pertain to the carrier concentration in the QW? Similarly, the text in lines 116 and 117 is unclear.]

Response 5: [Thank you for pointing those out. We agree with those comments. There are some errors of expression here. They're actually all injected carrier concentrations. So we've made a change to unify the names, and you can find them between lines 134 and 148.]

Comments 6: [Figure 2 presents the results of gain calculations. Firstly, the authors omit the effect of temperature, as inferred from the figure itself. Secondly, what losses do the authors assume when converting carrier concentration to gain? For example, how are losses represented in the ABC model? Unfortunately, the brief mention that the calculations are performed using the kp model provides no clarity.]

Response 6: [Thank you for pointing those out. We agree with those comments. Allow us to make the following explanation. First, we use PICS3D for band and gain calculations, and its internal logic is kp theory. To clarify this, we have improved the statements in lines 100 and 117. Secondly, the original intention of our design is to ensure the output power and efficiency of the device at room temperature, so the variable temperature is not considered. The suggestion you put forward is constructive and we think it can be used as a follow-up study. Third, in PICS3D, there are internal optical losses and mirror losses. The internal optical loss is coherent with the doping concentration (as shown in formula 2), and the doping values have been given in various tables in the manuscript in line 252. The mirror loss comes from the transmission of the cavity surface and is mainly affected by the reflectivity of the cavity surface. For calculations involving the overall loss of the EEL, we give the reflectance of the cavity surface in lines 173 and 255.]

Comments 7: [What exactly does the statement in lines 156 and 157 mean: “Using finite element analysis, optical field simulations were conducted on the designed 730 nm EEL …”? FEM is a numerical method for solving partial differential equations. Such a description does not clarify what the authors specifically did.]

Response 7: [Thank you for pointing this out. We agree with this comment. We have replaced “sing finite element analysis, optical field simulations were conducted on the designed 730 nm EEL” with  "simulated by COMSOL", in line 201.]

Comments 8: [Concerning Figures 4 and 5, the authors should describe the axes and labels on the figures more carefully. Abbreviations or concatenated words should not be used as labels unless their meaning is explained in the figure captions.]

Response 8: [Thank you for pointing those out. We agree with those comments. We added descriptions for the labels in Figure 3, 4 and 6, on lines 150, 192 and 243.]

Comments 9: [Comparing Figure 5, where gain is given in units of 1/cm, with Equation (1) and the text in lines 163 and 164, it seems there is an issue with the units. What do losses of 0.1 1/cm represent? Are these material losses or modal losses? The authors should create a table listing the individual laser layers, along with their absorption losses and refractive indices.]

Response 9: [Thank you for pointing those out. We agree with those comments. Please allow us to make the following statement. First of all, the unit of gain is 1/cm, the unit of optical loss coefficient is cm2, and the unit of loss is the same as the gain. Secondly, in the EEL, the threshold Gain is equal to the sum of the internal optical loss and the cavity surface loss, and only by overcoming these two losses can the device achieve lasing (Gain > 0 in Figure 3 (b)). The internal optical loss, which varies with the type of material and the dose of doping (given by formula 2), is the loss of light waves propagating in the ridged waveguide. The loss of cavity surface is caused by the transmission of front and rear cavity surface and is related to the reflectivity. When the length and reflectivity of the cavity are determined, the loss of the cavity surface is constant, and the total loss is only related to the internal optical loss. This is the premise of our waveguide design (Section 2.2). Third, we add Table 2 to give the refractive index of each material, while the internal optical loss of each layer is given by Figure 6.]

Comments 10: [What does the statement in lines 230 and 231 mean: “… a doping concentration five times the baseline value was selected for the final 230 design”? Unfortunately, the baseline value of the concentration does not appear to be provided in the work. This confirms the need for a table as mentioned in point 2.]

Response 10: [Thank you for pointing this out. We agree with this comment. We have revised the presentation of the data in Section 2.3 and clarified the base value for doping in line 252.]

Comments 11: [In line 236, it says Figure 1. This is an error. It should be Figure 6. Furthermore, the vertical axes on this figure lack labels. How do the losses shown in this figure relate to the losses mentioned earlier in the text? Are the loss values appropriate?]

Response 11: [Thank you for pointing those out. We agree with those comments. First, we changed the figure number on line 269 and corrected Figure 7. Second, the "Loss" in Figure 7 is the "internal optical loss". Thirdly, regarding the values problem, it can be seen from Figure 6 that the internal optical loss is mainly affected by the P and N waveguides, and according to formula 2, it is proportional to the doping concentration. When the doping concentration of P and N waveguides is increased and the optical field distribution is unchanged, the overall internal optical loss is basically proportional to the doping concentration.]

Comments 12: [In line 278, it says Figure 2. It should be Figure 7. Additionally, the figure caption should refer to both part (a) and part (b) of the figure.]

Response 12: [Thank you for pointing this out. We agree with this comment. We have changed the description of Figure 9.]

Comments 13: [Were the plots in Figure 10 generated for continuous wave (CW) operation or pulsed operation? Presumably, plots (a) and (b) were generated for CW operation. For what conditions was plot (c) generated? If it is also for CW operation, how did the authors estimate the wavelength shift with temperature? What can the authors say about the temperature in the laser’s active region during CW operation, emitting approximately 1 W of optical power?]

Response 13: [Thank you for pointing this out. Please allow us to explain. All of these figures were measured under CW conditions. We have added a description of "CW" in both Figure 9 and Figure 10. The wavelength redshift in Figure 12 (c) was not estimated, but measured by changing the laser heat sink temperature by setting the TEC temperature control module. Due to the measurement error, we used "approximately" in the expression.]

Reviewer 4 Report

Comments and Suggestions for Authors

The red semiconductor laser operating at a wavelength of 730 nm, known for its excellent monochromaticity, coherence and brightness, has found wide application. Compared with widely used light-emitting diode (LED) sources (LEDs), semiconductor lasers (SLs) have significant advantages, including narrow emission angle, excellent monochromaticity, strong coherence, high electro-optical conversion efficiency, and significant energy-saving advantages. However, quantum well (QW) materials commonly used at this wavelength often do not simultaneously meet the dual requirements of crystal lattice alignment and band gap alignment, which leads to a decrease in QW performance. In this study, a GaAsP/AlGaInP large-stress-compensated QW was developed. Stress compensation was developed to address the lattice mismatch while ensuring lasing action at 730 nm. Based on this, the influence of the waveguide design, in particular gradient and asymmetric waveguides, on the output power was studied. In this study, GaAsP was used as the material for the quantum well, while AlGaInP was applied on both sides as strain-compensating layers to reduce stress in the quantum well. The aim was to remove the compression stress that occurs between the QW and the GaAs substrates, thereby improving the overall quality of the QW region growth. At the initial stage of the study, the k·p theory was applied to calculate the QW gain spectrum at the same current density. The calculation results are illustrated by a number of figures. Using the k·p theory, energy levels and carrier distribution states in each layer were calculated. These results are also presented in a number of figures. To increase the carrier injection efficiency, the authors employed an AlGaAs gradient-separated confinement heterostructure (SCH) layer to provide additional potential energy for carrier injection, accelerate the transport efficiency of like-polarity carriers in the waveguide region, and suppress the transmission of opposite-polarity carriers on the same side. The epitaxial structure of the semiconductor laser designed in this study features a wide-strip ridge waveguide, enabling a 1 W single-bar output in a 730 nm edge-emitting laser. This study proposes an approach to enhance the lasing power and optoelectronic conversion efficiency of lasers and provide valuable solutions for their practical applications.

The article is written in clear and understandable language and undoubtedly deserves to be published in the journal Sensors.

Author Response

Dear reviewer,

Thank you very much for your recognition of our work.

However, we have made some revisions to this manuscript, mainly focusing on the summary of the existing research (Section 1) and the redescription of the calculation section (Section 2). At the same time, some tables and figures were added to illustrate the epitaxial structures of EELs.

These changes may show you more clearly what we do.

We sincerely hope to receive your suggestions and comments on this improved manuscript.

Wish you all the best and thanks again.

Sincerely,

Co-authors of the manuscript "sensors-3440557".

Reviewer 5 Report

Comments and Suggestions for Authors

The authors have written a very good paper on optimizing the design of a GaAsP/AlGaInP stress-compensated semiconductor red  laser at a wavelength of 730 nm. The optimization consisted in optimizing the waveguide design on which the laser operation is based. The article clearly identifies three directions in which this optimization was carried out, namely, a gradient-separated confinement heterostructure layer, asymmetric waveguide design, and doping optimization. The great advantage of the work is the practical implementation of an optimized laser design. Such an opimized laser achieved an output power of 1.5 W under a continuous driving current of 3 A. I believe that the work can be published in Sensors, but I have a few questions that I would like to get answered before publishing the article, here they are:

1. Is it possible to add a drawing with the general design of the laser under study in the article? 

It would make reading the article much easier. With P and N sides, input/output in waveguide etc.

2. Why do the authors use the expression "in foreign studies" on lines 51, 51? After all, the reader may not be from the same country with them.

3. Can you explain in more detail on lines 87, 88 why "a reduction in the P component is more favorable for stress release and material growth stability"?

4. Is it possible to make more explanations for the notation in the caption to the graphs in Fig. 2a? There are just inscriptions in the figure like EvL, EvH etc.

5. In Fig. 4, it is not clear exactly where the refractive index itself is?

6. It is not entirely clear what is the main advantage of the optimized 730 nm EEL proposed by the authors and the earlier version proposed by German colleagues, where the power is also > 1 W? This discussion is available in the Discussion section. Is it possible to describe this in more detail?

Author Response

Comments 1: [Is it possible to add a drawing with the general design of the laser under study in the article? It would make reading the article much easier. With P and N sides, input/output in waveguide etc.]

Response 1: [Thank you for pointing this out. We agree with this comment. We have added Figure 1 for all EELs in the article to compare the differences between them, and we have added Figure 8 to illustrate the structure of EELs.]

Comments 2: [Why do the authors use the expression "in foreign studies" on lines 51, 51? After all, the reader may not be from the same country with them.]

Response 2: [Thank you for pointing this out. We agree with this comment. We changed the expression between in line 46.]

Comments 3: [Can you explain in more detail on lines 87, 88 why "a reduction in the P component is more favorable for stress release and material growth stability"?]

Response 3: [Thank you for pointing this out. We agree with this comment. We changed “a reduction in the P component is more favorable for stress release and material growth stability” to “The reduction of P component in the well will reduce the lattice constant of GaAsP, which is more conducive to strain release and growth stability. However, it causes wavelength to redshift from 730nm as shown in Figure 2(f)” in line 124.]

Comments 4: [Is it possible to make more explanations for the notation in the caption to the graphs in Fig. 2a? There are just inscriptions in the figure like EvL, EvH etc.]

Response 4: [Thank you for pointing this out. We agree with this comment. We added interpretations of the notations in Figure 2(a).]

Comments 5: [In Fig. 4, it is not clear exactly where the refractive index itself is?]

Response 5: [Thank you for pointing this out. We agree with this comment. We modified Figure 5 to label the waveguide region and declare which EEL it belongs to. The epitaxial structure and refractive index values of the EEL are given in Figure 1 and Table 2 respectively.]

Comments 6: [It is not entirely clear what is the main advantage of the optimized 730 nm EEL proposed by the authors and the earlier version proposed by German colleagues, where the power is also > 1 W? This discussion is available in the Discussion section. Is it possible to describe this in more detail?]

Response 6: [Thank you for pointing this out. We agree with this comment. In the first part (lines 46 to 61), we summarize the progress of various parties and re-refine the significance of this study (lines 62 and 390). In this study, GaAsP/AlGaInP QW structure is proposed, and an asymmetric gradient SCH is used to achieve 730nm lasing, which provides a new technical scheme for this wavelength.]

Round 2

Reviewer 1 Report

Comments and Suggestions for Authors

The authors have revised the manuscript, resulting in significant improvements. However, several aspects still require further revision.

Incorrect interpretation of effects: The authors misinterpret some observed effects. 1) the effect described as “the number of carriers escaping the well through thermal excitation gradually approaches or even exceeds the number of carriers captured and participating in radiative recombination” may arise from a high carrier density in the well and is not directly related to the thickness of the well, 2) the phenomenon of “negative gain enhancement” is due to the need for a higher carrier density in a wider well to maintain gain, 3) the “relative positions of the conduction and valence bands” are not determined by the well thickness, 4) “higher-energy carriers” participate in radiative recombination not due to changes in the “relative positions of the conduction and valence bands,” but because wider wells introduce higher-energy levels that are filled by these higher-energy carriers, 5) the optical confinement factors do not directly influence output power (line 188).

The conclusion on line 318, stating “a power saturation due to carrier recombination saturation,” is unsubstantiated and should be removed from the sentence.

The authors provided an unsatisfactory response to Comment 10 [In Figure 2(a), it is evident that the electron and hole concentrations in the quantum well are not equal. How can the authors comment on this?]: Response 10: [Thank you for pointing this out. Please allow us to explain: because the effective mass of the hole is larger and the migration rate is lower, the carrier concentration of the hole in QW is usually lower.]. It should be noted that the electron and hole densities within the well must be equal to maintain charge neutrality.

In line 181, the term “injection efficiency” is not applicable in this context; I recommend removing this sentence. 

In Figure 10(a), please specify that the slow-axis far-field was calculated. Additionally, could the authors provide a more detailed description of how these fields were calculated were calculated and measured: angular resolution?

In Figure 12(c), please indicate the pump currents at which the spectra of EEL3 were measured.

On lines 166, 169 186 187 217 please replace “optical limiting factor” with “optical confinement factor”.

On lines 210, 212 please replace “optical loss coefficients” with “absorption cross-section”.

On line 104 please replace “material gain size” with “material gain”

On line 107 please replace “gain size” with “gain”

Comments on the Quality of English Language

Language Improvements: The English language in the manuscript could be further improved. For instance, phrases on lines 63 (“lattice distortion”), 74, 76, 77, 277 (“сompleted”), 75 (“done”), 125 (“conducive”), 138 (“distribution”), 168 (“achieve the concentration”), 182, 183 (“promotion”), 233 (“compressed”), 280 (“treated”), 284 (“stress release”), 365 (“power drops to approximately the same level as observed at room temperature (20 °C)”), 407 (“compressed”) could all benefit from refinement.

On line 104 please replace “material gain size” with “material gain”

On line 107 please replace “gain size” with “gain”

Author Response

Comments 1: [Incorrect interpretation of effects: The authors misinterpret some observed effects: (1) the effect described as “the number of carriers escaping the well through thermal excitation gradually approaches or even exceeds the number of carriers captured and participating in radiative recombination” may arise from a high carrier density in the well and is not directly related to the thickness of the well; (2) the phenomenon of “negative gain enhancement” is due to the need for a higher carrier density in a wider well to maintain gain; (3) the “relative positions of the conduction and valence bands” are not determined by the well thickness; (4) “higher-energy carriers” participate in radiative recombination not due to changes in the “relative positions of the conduction and valence bands,” but because wider wells introduce higher-energy levels that are filled by these higher-energy carriers; (5) the optical confinement factors do not directly influence output power (line 188).]

Response 1: [Thank you for pointing those out. We agree with those comments. Therefore, we made the following changes. First, we changed "As illustrated in Figure 2(a - e), the material gain size increases when the well thickness becomes thicker. This is because a thicker well can accommodate more carriers, resulting in a higher number of photons being generated by radiative recombination, thereby increasing the gain size. However, the well thickness cannot be infinitely increased. As the well becomes thicker, the quantum confinement effect on carriers weakens. Consequently, the number of carriers escaping the well through thermal excitation gradually approaches or even exceeds the number of carriers captured and participating in the radiative recombination. This results in a gradual reduction in the gain peak increment, ultimately leading to a negative gain enhancement. Weakening of the confinement also broadens the energy levels, directly affecting the relative positions of the conduction and valence bands. This not only results in a decrease in the photon energy produced from radiative recombination but also causes higher-energy carriers to participate in radiative recombination. Subsequently, this leads to a wavelength redshift and spectral broadening, as observed in the gain spectrum" to "As shown in Figure 2 (a-e), material gain increases with the thickness of the well. This is because thicker wells can hold more charge carriers, which increases the gain by producing more photons through radiation recombination. However, well thickness cannot increase indefinitely. As the well thickens, a higher carrier density is required to maintain the gain, which results in a decrease in the peak gain. Moreover, wider wells allow high-energy carriers to be filled to higher energy levels, resulting in wave-length redshifts and spectral broadening, as observed in gain spectrum", between lines 104 and 110. Second, we removed "which indicates that the QW of EEL1 has higher output power under the influence of graded SCH."]

Comments 2: [The conclusion on line 318, stating “a power saturation due to carrier recombination saturation,” is unsubstantiated and should be removed from the sentence.]

Response 2: [Thank you for pointing this out. We agree with this comment. Therefore, we had deleted "and carrier recombination saturation", on line 309.]

Comments 3: [The authors provided an unsatisfactory response to Comment 10 [In Figure 2(a), it is evident that the electron and hole concentrations in the quantum well are not equal. How can the authors comment on this?]: Response 10: [Thank you for pointing this out. Please allow us to explain: because the effective mass of the hole is larger and the migration rate is lower, the carrier concentration of the hole in QW is usually lower.]. It should be noted that the electron and hole densities within the well must be equal to maintain charge neutrality.]

Response 3: [Thank you for pointing this out. The data in Figure 3 (a) was calculated by PICS3D. The software uses SRH Model and Trap Model to solve the carrier concentration. Because the mobility of different types of carriers is different, their distribution is also affected. Please notice that, as shown in Figure 3 (a), although the carrier concentration of the hole in QW exceeds that of the electron, the electron concentration in the barrier especially in the P side exceeds the portion of the hole, which is due to the higher mobility of the electron. This causes the slight difference of the carrier densities within the quantum well when the device is electrically neutral as a whole.]

Comments 4: [In line 181, the term “injection efficiency” is not applicable in this context; I recommend removing this sentence.]

Response 4: [Thank you for pointing this out. We agree with this comment. Therefore, we had deleted "This demonstrates that carrier injection was more efficient in the graded SCH structure", on line 174.]

Comments 5: [In Figure 10(a), please specify that the slow-axis far-field was calculated. Additionally, could the authors provide a more detailed description of how these fields were calculated were calculated and measured: angular resolution?]

Response 5: [Thank you for pointing those out. We agree with those comments. Therefore, we added the description of the way in which the far-field divergence Angle and the beam quality factor were measured and calculated on lines 313 to 326.]

Comments 6: [In Figure 12(c), please indicate the pump currents at which the spectra of EEL3 were measured.]

Response 6: [Thank you for pointing this out. We agree with this comment. Therefore, we added "at 2A" on line 367.]

Comments 7: [On lines 166, 169 186 187 217 please replace “optical limiting factor” with “optical confinement factor”.]

Response 7: [Thank you for pointing those out. We agree with those comments. All " optical limiting factor" had been changed to " optical confinement factor " on lines 158,161,177,178,207.]

Comments 8: [On lines 210, 212 please replace “optical loss coefficients” with “absorption cross-section”.]

Response 8: [Thank you for pointing those out. We agree with those comments. All "optical loss coefficients" had been changed to "absorption cross-section" on lines 200 and 202.]

Comments 9: [On line 104 please replace “material gain size” with “material gain”.]

Response 9: [Thank you for pointing this out. We agree with this comment. The "material gain size" had been changed to "material gain" on line 104.]

Comments 10: [On line 107 please replace “gain size” with “gain”.]

Response 10: [Thank you for pointing this out. We agree with this comment. The "gain size" had been changed to "gain" on line 108.]

Comments 11: [Language Improvements: The English language in the manuscript could be further improved. For instance, phrases on lines 63 (“lattice distortion”), 74, 76, 77, 277 (“сompleted”), 75 (“done”), 125 (“conducive”), 138 (“distribution”), 168 (“achieve the concentration”), 182, 183 (“promotion”), 233 (“compressed”), 280 (“treated”), 284 (“stress release”), 365 (“power drops to approximately the same level as observed at room temperature (20 °C)”), 407 (“compressed”) could all benefit from refinement.]

Response 11: [Thank you for pointing those out. We agree with those comments. Therefore, we made the following changes. The "lattice distortion" had been changed to "lattice mismatch" on line 63; the "Completed in Section 2.1 was EEL0" had been changed to "EEL0 was designed in Section 2.1" on line 74; the "EEL2 was completed in Section 2.2.2" had been changed to "EEL2 was designed in Section 2.2.2" on line 76; the "which was completed in Section 2.3" had been changed to "which was designed in Section 2.3" on line 77; the "completed" had been changed to "includes" on line 267; the "Done in Section 2.2.1 was EEL1" had been changed to "EEL1 was designed in Section 2.2.1" on line 75; the "which is more conducive to strain release and growth stability" had been changed to "which reduces strain and increases growth stability "in line 118; the "distribution" had been changed to "concentration" on line 130; the "achieve the concentration" had been changed to "shrink" on line 160; the "promotion" had been changed to "increase" on lines 173 and 174; the "compressed" had been changed to "thinned" on line 223; the "treated" had been changed to "grown" on line 270; the "stress release" had been changed to "strain" on line 274; the “However, at 60 °C, the power drops to approximately the same level as observed at room temperature (20 °C)” had been deleted, on line 366; the "compress" had been changed to "shrink" on line 407.]

Reviewer 3 Report

Comments and Suggestions for Authors

I have no comments. The paper can be published in its current form.

Author Response

Dear reviewer,

Thank you very much for your recognition of our work.

However, we have made some revisions to this manuscript, mainly focusing on the active region design (Section 2.1) and the measurement of far-field divergence angle (Section 4). At the same time, some language mistakes have been changed.

These changes may show you more clearly what we do.

We sincerely hope to receive your suggestions and comments on this improved manuscript.

Wish you all the best and thanks again.

Sincerely,

Co-authors of the manuscript "sensors-3440557".

Reviewer 5 Report

Comments and Suggestions for Authors

I think the paper can be published in present form.

Author Response

(The authors gave the same response as above.)

Round 3

Reviewer 1 Report

Comments and Suggestions for Authors

The authors have revised the manuscript with significant improvements. The manuscript focuses on the development of semiconductor lasers operating at a wavelength of 730 nm based on GaAsP/AlGaInP/AlGaAs/GaAs heterostructure, which is a unique feature of the study.